# Development of an Immunofluorescent Capillary Sensor for the Detection of Zearalenone Mycotoxin

**DOI:** 10.3390/toxins14120866

**Published:** 2022-12-09

**Authors:** Krisztina Majer-Baranyi, Attila Barócsi, Patrik Gádoros, László Kocsányi, András Székács, Nóra Adányi

**Affiliations:** 1Food Science Research Group, Institute of Food Science and Technology, Hungarian University of Agriculture and Life Sciences, Villányi út 29-43, H-1118 Budapest, Hungary; 2Department of Atomic Physics, Institute of Physics, Budapest University of Technology and Economics, Műegyetem rkp. 3, H-1111 Budapest, Hungary; 3Agro-Environmental Research Centre, Institute of Environmental Sciences, Hungarian University of Agriculture and Life Sciences, Herman Ottó út 15, H-1022 Budapest, Hungary

**Keywords:** mycotoxin, zearalenone, immunosensor, immunofluorescence, capillary biosensor

## Abstract

A capillary-based immunofluorescence sensor was developed and incorporated in a flow injection analysis system. The light-guiding capillary was illuminated axially by a 473 nm/5 mW solid state laser through a tailored optofluidic connector. High sensitivity of the system was achieved by efficiently collecting and detecting the non-guided fluorescence signal scattered out along the wall of the capillary. The excitation was highly suppressed with bandpass and dichroic filters by simultaneously exploiting the guiding effect inside the capillary. The glass capillary used as a measuring cell was silanized in liquid phase by 3-aminopropyltriethoxysilane (APTS), and the biomolecules were immobilized using glutaraldehyde inside the capillary. The applicability of the developed system was tested with a bovine serum albumin (BSA)—anti-BSA-IgG model-molecule pair, using a fluorescently labeled secondary antibody. Based on the results of the BSA–anti-BSA experiments, a similar setup using a primary antibody specific for zearalenone (ZON) was established, and a competitive fluorescence measurement system was developed for quantitative determination of ZON. For the measurements, 20 µg/mL ZON-BSA conjugate was immobilized in the capillary, and a 1:2500 dilution of the primary antibody stock solution and a 2 µg/mL secondary antibody solution were set. The developed capillary-based immunosensor allowed a limit of detection (LOD) of 0.003 ng/mL and a limit of quantification (LOQ) of 0.007 ng/mL for ZON in the competitive immunosensor setup, with a dynamic detection range of 0.01–10 ng/mL ZON concentrations.

## 1. Introduction

Mycotoxins are secondary metabolites of filamentous fungi, which contaminate cereals and foodstuffs worldwide, causing vast agricultural productivity and economic losses. Numerous mycotoxins exist, but among them, approximately 500 are reported as potentially toxic [1]. Fusarium fungi are the most widespread in cereal-growing areas in the temperate climate zone. They contaminate practically all food and feed crops, including corn, barley, oats, wheat, sorghum, millet, rice, flour, malt, and soybeans [2,3]. Zearalenone (ZON) is a fusarial toxin primarily produced by *Fusarium graminearum* and *F. culmorum*, but also biosynthetized by *F. cerealis*, *F. equiseti, F. semitectum*, and *F. moniliforme*. The toxin is produced at low temperatures (below 25 °C) and high humidity in the field during harvest or in the course of storage [4,5]. The formation of the toxin during storage can be prevented by proper drying of the harvested plant. ZON appears as a surface pollutant on cereals, and a significant part of it can be removed with bran after mill processing [6]. Moreover, it is also an emerging water pollutant [7].

ZON is considered a xenoestrogen. Due to the similarity of its chemical structure to natural estrogens (such as 17β-estradiol), it can bind to estrogen receptors, stimulating expression of estrogen-responsive genes in a number of animal species, especially in pigs, leading to amplified estrogenicity and consequent reduced fertility [8]. In addition, ZON has immunotoxic, hepatotoxic, hematotoxic, and nephrotoxic effects and also enhances lipid peroxidation [9,10]. The International Agency for Research on Cancer (IARC) classified ZON as a Group 3 carcinogen (unclassifiable as to carcinogenicity in humans) due to inadequate evidence for human carcinogenicity [11]. Due to its proven endocrine disruptive effects, the EU has established maximum permitted levels for ZON in certain foodstuffs in a range of 20 μg/kg (for processed cereal-based foods and baby foods for infants) and 350 μg/kg (for unprocessed maize) [12]. To follow these regulations, comprehensive detection techniques are required to monitor ZON contamination in food and feed.

Despite the high sensitivity and accuracy of conventional analytical methods—such as high-performance liquid chromatography (HPLC) hyphenated with different detectors [13,14,15,16], liquid chromatography coupled with mass spectrometry (LC-MS), liquid chromatography-tandem mass spectrometry (LC-MS/MS) [17], and gas chromatography-tandem mass spectrometry (GC-MS/MS) [18]—they suffer from several limitations, including involvement of complicated instrumentation or being inexpedient for on-site application. Therefore, several alternative analytical techniques have emerged in recent decades. Among them, immunoassay methods—such as commercially available enzyme-linked immunosorbent assay (ELISA), lateral flow immunoassays (LFA), and flow-through immunoassays—have become most popular. Although they are suitable for large-scale sample screening and detection with low cost, they still have limitations. In the case of ELISA, the assay requires incubation periods between 1 and 2 h. In the case of LFA and flow-through tests, their sensitivity falls short of that of traditionally used techniques, despite their ability to yield qualitative and/or semi-quantitative results [19]. To overcome these limitations, biosensors, including immunosensors, could be the most promising tools that can meet most of the requirements to become efficient analytical devices for on-site mycotoxin determination [20]. There are several types of biosensors for mycotoxin monitoring based on the applied transducer, but optical biosensors, particularly fluorescent immunosensors, have become one of the most rapidly emerging techniques due to their exceptional sensitivity and simplicity. In addition to the formerly applied organic dyes and fluorescent proteins, quantum dots (QDs) [21,22,23,24], quantum nanobeads (QBs), fluorescent metallic nanoparticles (MNPs), upconversion nanoparticles (UCNPs) [25,26,27], and time-resolved fluorescence materials (TRFMs) [28] are reported as fluorescent probes [29,30,31] as they provide unique chemical and optical properties, such as high photostability, bright fluorescence, and good biocompatibility. Fluorescence detection enables high detection sensitivity (such as single molecule detection) and provides fast response times. Its further advantages are localized fluorescence signal, multiplexed assays using multicolor dyes, and a straightforward labeling process, which provides appropriate functional groups on the target [32].

In this study, a capillary-based fluorescent immunosensor was developed for detection of ZON. The main objective was to develop a simple, low-cost device and assay platform for initial screening of ZON contamination in agricultural products, either in smaller labs or onsite, allowing analysis even by unskilled personnel prior to intensive laboratory testing. A competitive assay format was used for ZON determination, where the immobilized ZON-BSA conjugate competed with the ZON present in the sample for the anti-ZON antibodies in the aqueous phase. The level of antibodies bound to the immobilized ZON-BSA conjugate was determined by the addition of immunoglobulin-specific secondary antibodies labeled with a green fluorescent dye, CF^TM^488A. Therefore, concentration-dependent decreases in the fluorescent signal could be observed. The effectiveness of the immunosensor was evaluated using spiked samples. Altogether, this method is easy to use and rapid for simple analysis of ZON.

## 2. Results and Discussion

### 2.1. Capillary Waveguide-Based Biosensor Design

The operating principle of the biosensor is based on a glass capillary acting as an optical waveguide once filled with a sample solution (Figure 1). Axially coupling a laser beam into the filled-up capillary will excite the labeling fluorophores at high power density. A large contrast of the low photon count fluorescent response to the excitation can be maintained because fluorescence is scattered over the full solid angle, whereas excitation will be strongly guided, hence confined, in the capillary. The combination of a reflecting parabolic mirror and a collimating lens efficiently collects (from a large solid angle) and collimates the fluorescence signal scattered through the mantle of the capillary (Figure 2a). The residual excitation signal scattered out from the capillary is efficiently reduced by applying a fluorescence emission filter and a dichroic filter. A large area photodetector collects the fluorescence from the clear aperture of the 25 mm diameter optics.

To ensure a temporally stable environment, the immunofluorescent sensor was operated in a continuously flowing injection analysis (FIA) system consisting of a flow-through capillary and a sample injector with a sample loop of 200 µL. A constant flow was maintained inside the capillary sensor by a peristaltic pump through a specially designed optofluidic connector (Figure 2b). The sample solution was loaded into the sample loop of a manual sample injector, which then switched to inject the sample into the capillary without flow disruption. The constant flow approach allows the temporal recording of fluorescence rise and fall, thus further improving the sensitivity of the system, as correlated sampling can be employed by subtracting the background signal, preceding the sample influx, from the fluorescence signal.

In the immunofluorescent sensor setup described above, the antigen or its conjugate is fixed in the capillary, similar to the planar immunosensors [33,34]. The immunoreactions completed by the proper antibody are detected by the reaction of the second antibody labeled by a fluorescent dye. Anti-IgG–CF^TM^488A (a-IgG–CF488 i.e., anti-rabbit IgG (H + L) antibody produced in F(ab’)2 fragment of goat, λ_ex_ = 490 nm; λ_em_ = 515 nm; conjugated to CF™488A) was used to signal the immunocomplex formed during the immune reaction, where the conjugated CF^TM^488A dye yielded a fluorescence signal corresponding to the concentration of the primary antibody bound to the immobilized antigen conjugate during the immunoreaction.

### 2.2. Protein Adsorption on the Inner Surface of the Silanized Capillary

To investigate the analytical application of the new instrument construction and to develop the immobilization and measuring method, we used a pair of model molecules, namely BSA–a-BSA, because it models the conjugate (usually BSA) and antibody used for selective measurements well. First, the silanized capillaries and the capacity of the layers were examined by protein adsorption. During the measurement, an aqueous solution of a-IgG–CF488A secondary antibody conjugated to a fluorescent dye at a concentration of 1 µg/mL was used. Next, the baseline was determined with distilled water, then the antibody-dye conjugate solution was injected into the measuring cell. For the measurement, a 0.12 mL/min flow rate was used. The degree of surface adsorption was measured after washing out the unabsorbed sample and stabilization of the baseline (9 min after injection). In all cases, only minimal adsorption of the dye-labeled secondary antibody was observed on the surface, yielding a signal level of 3–4 relative fluorescence units (RFUs). When anti-BSA (a-BSA, 10 µg/mL) was first passed through the measuring cell, followed with the injection of a-IgG–CF488A solution, the signal showed only a slight increase (7–9 RFUs), indicating that the dye-conjugated antibody bound to the a-BSA antibodies adsorbed onto the capillary.

### 2.3. Coupling Potential of Protein Molecules onto the Inner Surface of the Silanized Capillary

The functional—most often amino—groups applied to the surface by silanization can be converted into other reactive groups by chemical reactions or derivatization in one or more steps, to which the protein type compounds can be attached. In our silanization experiments, the formed layers were evaluated by examining an antigen–antibody model-molecule pair, BSA—a-BSA-IgG, respectively.

In order to monitor the immobilization and the biochemical reaction, first the effectiveness of a-IgG–CF488A dye conjugate was tested in the immobilization steps also. For the immobilization, a 2.5% (d.w.) glutaraldehyde (GA) solution was injected applying a 0.08 mL/min flow rate. This bivalent aldehyde bounded to the amino groups formed on the inner surface of the silanized capillary and also covalently attached the protein-type molecules (20 µg/mL BSA in this case) to the surface by their amino groups. The injected GA solution gave a large negative signal in all cases, presumably due to its optical density, but the signal returned to baseline when the molecules that remained unattached were washed out of the measuring capillary. After washing, the flow of distilled water was changed to TRIS buffer (42 mM/L, pH 7.4). That was followed by injection and immobilization of a-BSA (10 µg/mL in TRIS buffer). The a-BSA antibodies were allowed to undergo immunocomplex formation with the immobilized BSA molecules. After this complex formation, another washing step with 100 mM hydrochloric acid (HCl) was carried out for 10 min. Next, the flow rate was changed to 0.12 mL/min, different concentrations of a-IgG–CF488A dye conjugate (1, 2.5 and 5 µg/mL) were injected, and the magnitude of the fluorescence signals was examined (Figure 3) through the mantle of the a-BSA-sensitized sensor. After each injection, the protein complex was washed out of the capillary with 100 mM HCl to regenerate the surface. The regeneration took only 5 min after each injection. The end point of the measurement was considered the data between 9–12 min, and the average of these data was used for the evaluation. As the concentration of the dye increased (1, 2.5, and 5 µg/mL), the magnitude of the corresponding signal also increased by an average of 17.6, 31.2, and 88.6 RFUs, respectively, proving that the dye is a suitable indicator for measuring the a-BSA antibody.

In indirect immunosensors, the antigen or, in case of small antigen molecules, antigen conjugate has to be immobilized. Therefore, BSA (20 µg/mL) was covalently bound in the silanized capillaries by GA. The spontaneous adsorption of the dye was recorded to check whether the a-IgG–CF488A dye conjugate at a concentration of 1 μg/mL gave a signal after washing out the unabsorbed molecules. The dye conjugate injected onto the fixed BSA gave a very small signal (Figure 4; A,B: 5.2 ± 0.6 RFUs), indicating that the dye-conjugated secondary (anti-IgG) antibody adsorbed weakly to the immobilized BSA. Evaluable signals for the dye were obtained only after the injection of a-BSA (10 µg/mL). The results were examined by injecting the antibody separately, followed by the injection of the dye conjugate or by mixing the antibody with the dye conjugate before injection. It was found that the signal increased with the injection of the a-BSA (Figure 4; C: 22.6 ± 0.5 RFUs), indicating that the dye-conjugated secondary antibody bound to the a-BSA that bound to the immobilized BSA molecules. A better signal was obtained in the case of injecting the mixture of the antibody and the a-IgG–CF488A dye conjugate (Figure 4; D: 33.9 ± 1,0 RFUs) than injecting separately. The incubation of the above-mentioned mixture before injection did not give a different result.

Antibody concentration is one of the key parameters in competitive immunoreactions affecting the sensitivity of the sensor. After immobilizing BSA (20 µg/mL), different concentrations of a-BSA were injected (2 µg/mL and 10 µg/mL, respectively) to check how the primary antibody concentration affects the signal response. The a-BSA sample was injected into the capillary in two different ways: (1) the antibody was injected first and the dye conjugate separately and (2) the mixture of the dye and the antibody was applied. The recorded signals were 4.2 ± 0.5, 10.8 ± 0.7; 28.3 ± 0.6, and 35. 9 ± 0.8 RFUs for BSA only, for 2 µg/mL a-BSA and 10 µg/mL a-BSA injected separately from the dye, and for 10 µg/mL a-BSA mixed with the dye, respectively (Figure 5).

### 2.4. Effect of Acetonitrile

As an organic solvent, usually acetonitrile (AcN), was used to extract the mycotoxins from the samples for the further applications, the effect of AcN on the immune response, as well as on the fluorescent detection, was investigated (Figure 6). The potential interfering effect may affect the protein-type antibodies or the fluorescence measurement and signal, not specifically in the determination of the tested mycotoxin. Therefore, we also examined this effect in the BSA–a-BSA system in order to make a valid conclusion for further applications. Samples of a-BSA were pretreated in an AcN:water (6:4) mixture and diluted with TRIS buffer before the measurement. Different rates of dilution of the extracting solution were tested to check if AcN affects the immunoreaction or the fluorescent signal. At a 1:1 mixture of a-BSA antibody dissolved in TRIS buffer and extracting solution mixed with the solution of a-IgG–CF488A, a significant decrease of the signal was observed (13.9 ± 0.8 RFUs) compared to previous results, indicating that AcN affects the signal of the proper biochemical reactions and also the fluorescent detection (Figure 6; A). Using 25% (Figure 6; B) or even 10% (Figure 6; C) of AcN mixture, the response signal increased up to 41.0 ± 1.3 and 43.0 ± 1.16 RFUs, respectively. We found that AcN below 10% does not interfere with the antibody solution used in competitive measurements (a-BSA or in further measurements of the a-ZON) in the required diluting range; for further measurements, 100× dilution was used.

### 2.5. Fluorescence-Based Competitive Immunosensor for Measuring ZON

For the measurement of ZON, a competitive immunosensor was developed with capillary fluorescence detection in a FIA system, based on the described method for the BSA–a-BSA determination. In the competitive measurements, the sensitivity and stability of the sensor is affected by both the concentration of the antigen conjugate immobilized and the concentration of the antibody applied. Therefore, optimization of these parameters is crucial.

Comparing the responses of several capillary sensors sensitized with different concentrations of ZON-conjugate to sera, we found that increasing the concentration of the conjugate used for immobilization decreased the magnitude of the corresponding signal. During the fixation of 2 µg/mL ZON-BSA conjugate, the responses of the sensors were 20% higher than those of 20 µg/mL conjugate-sensitized sensors. However, in the latter case, the signals were more stable during repeated injections; thus, this concentration was used thereafter.

As discussed, in competitive measurements, sensitivity is affected by the antibody concentration. Therefore, upon immobilization of the ZON-BSA conjugate (20 µg/mL) on the inner capillary surface, sensor responses to polyclonal serum were studied at different dilutions of 50,000× 5000×, 2000×, 1000×, and 500× (Figure 7). Standard or sample solutions were mixed with antibodies at appropriate concentrations (diluted in TRIS (42 mM/L, pH 7.4)) in 1:1 ratio. The mixture was incubated under controlled circumstances (3 min at 31 °C) and was subsequently injected into the flow system. Upon incubation, only antibodies remaining in free form in the mixture can bind to the antigens immobilized on the sensor surface. Therefore, the amount of antibodies bound to the surface was inversely proportional to the antigen concentration in the standard solution. In terms of sensor response and sensitivity, the lower the amount of the antibody applied, the higher the sensitivity achieved, even at the cost of a reduced signal. With higher antibody concentrations, larger signals are obtained, but the sensitivity of the test is reduced. At high serum dilutions, the sensor became extremely sensitive. The majority of the antibody molecules were bound at higher antigen concentrations, but the sensor response became small and uncertain. Although the sensor response is sufficiently high at low serum dilutions, the sensitivity of the sensor deteriorates and only becomes sensitive at higher antigen concentrations. To set the operating conditions of the immunosensor, the serum dilution should be chosen such that the antibodies are just saturating the sensor surface. Figure 8 depicts the signal curves of the ZON standards.

From the artificially contaminated (spiked) samples, an AcN-water extract was prepared as described in Section 4.5, and the samples were further diluted with TRIS buffer (42 mM/L, pH 7.4) to diminish the effect of AcN (see in Section 2.4) and to determine the ZON concentration that still had appropriate sensitivity. A series of artificially contaminated barley samples were used to record the calibration curve.

Taking into consideration the height and shape of the signals, and the rate of inhibition measured for standard solutions in the case of different antibody concentrations, the optimal dilution of the polyclonal antibody stock solution was chosen to be 2500×. Using this working dilution, calibration curves were recorded for standard solutions and for the series of artificially contaminated barley samples, as well. As the matrix effect must be taken into account when examining spiked barley samples, the measured signals are significantly smaller than in the case of the standards. For the control standard and barley sample (containing no ZON), the signals were 57.4 ± 1.9 and 27.8 ± 0.5 RFUs, respectively. At the same time, if we plot the calibration curves by calculating the degree of inhibition, we obtain a curve with a similar course, and the statistical parameters are also similar (Figure 9). The measuring range was obtained between 0.01 and 10 ng/mL, while the LOD was 0.003 ng/mL and the LOQ was 0.007 ng/mL. The mean inhibition (EC_50_) values of the standard solutions and of the barley samples measured by the novel immunofluorescent capillary sensor was 0.68 ± 0.08 ng/mL and 0.83 ± 0.12 ng/mL, respectively. The EC_50_ value of the corresponding enzyme-linked immunosorbent assay (ELISA) measurement [35] was 2.73 ± 0.35 and 2.20 ± 0.31 ng/mL, using absorbance or fluorescence detection, respectively. The comparison was established on the basis of the ELISA measurement, because the sensitivity of the new instrument was questionable when using the same conjugate and antibody.

### 2.6. Investigation of the Selectivity of a ZON-Specific Sensor

We investigated the cross-reaction of various ZON derivatives and intermediates by the developed sensor. Taking the sensor response for ZON as 100%, we investigated how large of a signal the different derivatives give at the same concentration as compared to the ZON signal. ZON, α-zearalenol, α-zearalanol, and β-zearalanol were measured both with the novel immunosensor and with the ELISA reference method, using the same anti-sera. The results were 100%, 32.4%, 10.7%, and 3.2%, while for ELISA [35] they were 100%, 28.2%, 7.1%, and 1.1%, respectively. Based on the results, it was found that ZON can be selectively determined among the tested derivatives and there is no significant difference between the results measured with the immunosensor-based fluorescence detection and the data measured with the ELISA reference method; thus, the selectivity of the two methods is very similar.

## 3. Conclusions

A novel capillary-based fluorescent immunosensor was developed for the detection of ZON. This measuring setup is a simple device and assay platform for initial screening of ZON contamination in agricultural products, either in smaller labs or via onsite analysis conducted even by unskilled personnel, prior to intensive laboratory testing. Competitive measurements were performed for ZON determination. The concentration of antibodies linked to the immobilized ZON-BSA conjugate was determined by the addition of ZON-specific antibodies as a primary antibody, followed by secondary (IgG-specific) antibodies conjugated to a green fluorescent dye (CF^TM^488A). Therefore, concentration-dependent decreases in the fluorescent signal could be observed. Based on the results, the mean inhibition (EC_50_) values of the standard solutions and of the barley samples measured by the novel immunofluorescent capillary sensor were found to be 0.68 ± 0.08 ng/mL and 0.83 ± 0.12 ng/mL, respectively, while that for the ELISA measurement was 2.20 ± 0.31 ng/mL; thus, the selectivity of the two methods is also very similar. With the developed sensor, ZON content of 15 samples could be determined, where the measurement of one sample took 20 min.

## 4. Materials and Methods

### 4.1. Materials

Organic chemicals and solvents, (3-aminopropyl) triethoxysilane (APTS), mycotoxin ZON and its derivatives, BSA, BSA-specific polyclonal antibody, Anti-Rabbit IgG (H + L), CF™488A antibody produced in F(ab’)2 fragment of goat as secondary antibody, salts for buffers, and other chemicals were purchased from Sigma-Aldrich, Ltd. (Budapest, Hungary), unless otherwise stated. The purity of standard solutions was ≥98%.

### 4.2. Instrumentation

The complete measuring system is shown in Figure 10a. The sample solution flows through the chemically activated glass capillary (5 μL Accupette Pipets, calibrated TC, Type II glass E-438-71, Dade Diagnostics Inc., West Monroe, LA, USA) with 0.29 mm internal and 1.2 mm external diameters acting as a biosensor. The capillary with an O-ring on its end is inserted into the axial output port of the brass-made optofludic connector and pushed through the axially located holes of the 3D-printed housing. The optofluidic connector is then fixed into the nut of the housing with standard M10 × 1 threading to allow easy capillary replacement. When the thread is tightened, the O-ring forms a seal between the capillary mantle and the end-face of the connector body. The length of the capillary extends the housing so that a waste tube can be mounted.

The output of a 473 nm, 5 mW diode pumped solid state laser (4303-005-195, Uniphase) is coupled into a polymethyl-methacrylate core multimode optical fiber with core diameter of 1 mm (NT02-534, Edmund Optics Ltd., Barnington, NJ, USA) through an SMA905 fiber collimator (F220SMA-A, Thorlabs Inc., Newton, NJ, USA). The SMA905 output of the fiber is connected to the axial port of the optofludic connector. The internal hole of the connector is L-shaped to eliminate the formation of dead volume. The hole diameter is 0.9 mm to ensure that it fits within the fiber diameter. This design allows efficient optical coupling of the fiber–hollow waveguides, as well as the formation of a seal of the fluidic flow at the fiber end-face. Nevertheless, the threading of the SMA905 connector is properly sealed with a Teflon strip.

The input optical power axially coupled into the capillary is limited below 1 mW to avoid early fading of the fluorophore due to the high intensity experienced. When set to 0.5 mW measured at the output port of the optofluidic connector, the waveguiding effect of the capillary filled up with distilled water results in 0.42 mW (that is >80%) optical power measured at the output of the capillary, in contrast to the value of 0.26 mW when the capillary is empty. Notably, only the fraction of the input power propagating in the liquid-filled core region will excite the fluorophores. In addition to coupling loss, the transmission loss through the capillary is accounted for by the scattering losses from the liquid core, as well as the volume and surface scattering centers of the capillary material.

The side port of the optofluidic connector, that is, the sample solution inlet, is connected to the column output of the manual sample injector (7725, Rheodyne). The pump input of the injector is connected to a low-pulse peristaltic pump (Minipuls 3, Gilson Inc., Middleton, WI, USA).

The fluorescence signal emitted by the fluorophores in the core liquid and scattered through the mantle of the capillary is measured with a large active area (Ø27.9 mm) silicon photodiode (PIN-25D, OSI Optoelectronics, Hawthorne, CA, USA) through a 25.4 mm diameter collimating plano-convex lens (49-863, Edmund Optics Ltd., Barnington, NJ, USA) forming an axial beam path inside the measuring head (Figure 10b). Spectral blocking of the residual excitation signal is achieved by a combination of bandpass (FF01-531/40-25, Semrock, IDEX Health & Science, West Henrietta, NY, USA, peak: 531 nm, width: 40 nm) and dichroic (FD1G, Thorlabs, Inc., Newton, NJ, USA, low edge: 505 nm, high edge: 575 nm) filters. To minimize large angle scattering that reduces blocking of the filters and to increase the solid angle of photon collection, a collimating mirror is formed by bonding a reflective sheet (Silverlux XT-0008-2439-8, 3M Co., Saint Paul, MN, USA) onto the parabolic-shaped 3D-printed base of the capillary housing.

The photodetector is connected to an electrometer transimpedance preamplifier (OPA129, Texas Instruments, Dallas, TX, USA), output of which is fed to a variable gain instrumentation amplifier (AD620, Analog Devices, Cambridge, MA, USA). Gain and offset of the second stage can be controlled by a 4-channel, 8-bit (256-position) digital potentiometer (AD5263-50, Analog Devices, Cambridge, MA, USA). The conditioned and amplified signal is digitized with a 12-bit analog-to-digital converter (AD7864-2, Analog Devices, Cambridge, MA, USA) within the 0 to 2.5 V unipolar input range, yielding 4096 RFUs. A single-board computer (CMD16686GX, Real Time Devices Ltd., Wickwar, UK) controls the amplifier and digitizer units, and transmits the data via USB connection to a computer running the control software with a graphical user interface.

### 4.3. Immunogen and Antibody Production

ZON is a non-immunogenic substance due to its low molecular weight (C_18_H_22_O_5_, 318.36 g/mol). Therefore, it should be conjugated to a protein carrier for immunization. Considering that ZON does not have a functional group with which it could connect directly to a protein, it was reacted with 2-aminooxyacetic acid on the short oxime side chain to form a carboxyl group in the molecule [35,36] by the method of Thouvenot and Morfin [37]. The ZON derivative as hapten was conjugated to BSA, which was used for the immunosensor development, and also to conalbumin (ConA) as a carrier protein for the immunization. The polyclonal antibody against ZON was produced in female New Zealand white rabbits by injecting 25 μg of ZON-ConA conjugate per kg body weight and 50 μL of Freund’s complete or incomplete adjuvant periodically on days 1, 14, 28, and 42 in the thicker part of the skin above the scapula. Ten days after the last booster dose, blood collected from the marginal vein was centrifuged at 2400× *g* for 15 min. The obtained sera were purified for IgG by sodium-sulphate precipitation, according to the method of Harboe and Inglid [38].

### 4.4. Functionalization of the Inner Surface of the Capillary

Since the hydroxyl groups on the inner surface of the capillary are not suitable for direct binding/coupling of biomolecules, the surface must be chemically modified [39]. By silanization, various functional groups can be applied to the glass surface, for which the protein-type molecules bearing the amino group can be fixed by appropriate chemical steps. The most commonly used compound in both aqueous and organic phase silanization is APTS, which can be used to apply amino groups to the surface of the capillary. Silanization experiments were performed using the APTS trifunctional silane compound in aqueous phase using a continuous flow system.

The capillary was cleaned by NoCromix (3.6 g/100 mL; Godax Laboratories Inc., Cabin John, MD, USA) cleaning solution beforehand, then the surface was hydrated by hot distilled water (100 °C) for 1 h. Silanization was performed by 10% APTS solution (pH 3.0 adjusted with 1 mol/L HCl) at 75 °C for 3 h. Then, the capillary was washed with distilled water and heat-treated at 100 °C for 1 night. Silanization in the aqueous phase gave a well-reproducible, smooth, homogeneous, well-wettable layer. Based on previous results, under these parameters, silanization resulted in a silane layer with a thickness of approximately 1.3–1.7 nm, which did not affect the flow conditions [40]. Wet silanization is an inexpensive, well-reproducible method suitable for laboratory use, and sensors can be stored for 2–4 weeks. The silanized capillaries were stored at room temperature until use.

For the immobilization of the biomolecules, the aminosilanized surface was activated with 200 μL of 2.5% GA in distilled water. The immobilization was carried out in a FIA system using 0.08 mL/min flow rate without stopping at room temperature. After activation, the medium was changed to 42 mM TRIS buffer (pH 7.4), and 20 μg/mL ZON-BSA conjugate was injected into the system. Non-bound or excess molecules were washed off by 200 μL of 100 mM HCl. Afterward, the flow rate was changed for 0.12 mL/min, and the measurements were carried out at room temperature at this elevated flow rate condition. The sensor surface was regenerated by 100 mM HCl after each sample during measurement.

### 4.5. Sample Preparation and Measurement Conditions

During the test, uncontaminated barley seeds were ground with a Labmill grinder using a 1/60 sieve size, and the resulting grit was artificially contaminated with different amounts of ZON standard in the concentration range of 0.01–1000 µg/kg. Accurately weighed 1 g of cereal was mixed with 5 mL of AcN-water (60:40) for 30 min, then allowed to stand for 10 min, and the supernatant was centrifuged on an ultrafilter (50,000 NMWL) at 5000 rpm for 20 min to give the filtrates stored at 4 °C until use.

For the measurement, sample solutions were mixed with primary antibody diluted 2500 times (diluted in TRIS (42 mM/L, pH 7.4)) in 1:1 ratio. The mixture was incubated for 3 min at 31 °C and was subsequently injected into the flow system. An injector with a 200 µL loop and a flow rate of 0.12 mL/min was used for the measurement. After 5 min, 2 µg/mL secondary antibody solution was injected to signal the immunocomplex formed during the immune reaction. The data collected between the ninth and twelfth minutes after the injection were used for evaluation. It took 18–20 min to measure a sample, and the regeneration with 100 mM HCl took 5 min after each sample.

### 4.6. Enzyme-Linked Immunosorbent Assay for Zeareleone Determination

ELISA determinations were performed in 96-well microplates. A BSA conjugate of ZON at a concentration of 1 µg/mL in 0.1 M carbonate-bicarbonate buffer (15 mM Na_2_CO_3_, 35 mM NaHCO_3_, pH 9.6) was used as plate-coating antigens. After washing and blocking with 1% gelatine solution in phosphate buffer saline (PBS, 137 mM NaCl, 2.7 KCl, 10 mM Na_2_HPO_4_ × 2H_2_O, pH 7.4), samples or standard solutions and the antiserum diluted in PBS containing 0.05% Tween20 (pH 7.4) were dispensed into the wells and incubated. After another wash, bound antibodies were exposed to goat anti-rabbit IgG conjugated to horseradish peroxidase (IgG-HRP) (1:3000 dilution in PBST), and enzymatic activity was measured using 0.01 M hydrogen-peroxide as a substrate and 3 mM o-phenylenediamine as a chromophore in 0.5 M citrate buffer (pH 5.0). For standard curves, a stock solution of ZON in methanol (1 mg/mL) was used in serial dilution from 0.004 ng/mL to 2 µg/mL. Standard curves were calculated from the raw data using sigmoidal curve fitting. The detection limit was defined as the lowest concentration of the target analyte showing a reduction of 3 standard deviations from the mean blank standard absorbance.

### 4.7. Statistical Evaluation

All experiments were repeated three times, and all the data in the results were the average values of the measured data, which were processed by Microsoft Office Excel. Calibration curves were drawn by OriginPro 7 software (OriginLab Corp.) Sensitivity was determined by the limit of detection (LOD) and limit of quantification (LOQ). LOD is calculated S/N > 3, while LOQ is based on S/N > 10.

## Figures and Tables

**Figure 1 toxins-14-00866-f001:**
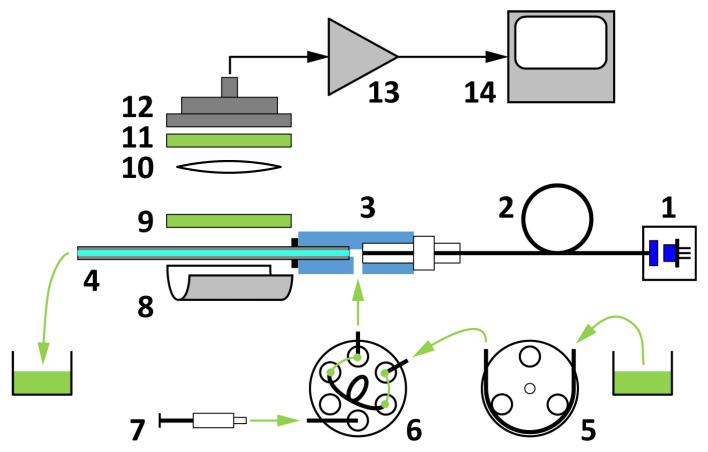
Block scheme of the capillary-based sensor setup: laser source (1), multimode optical fiber (2), optofluidic connector (3), glass capillary (4), peristaltic pump (5), manual sample injector (6), sample syringe (7), parabolic mirror (8), fluorescence emission filter (9), collimating lens (10), dichroic filter (11), large area photodetector (12), detector amplifier (13), and computer-connected signal processing unit (14).

**Figure 2 toxins-14-00866-f002:**
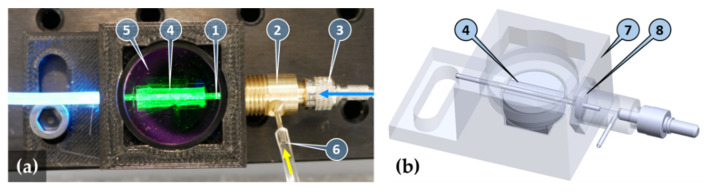
Image (**a**) and transparent view (**b**) of the sensor unit and the optofluidic connector: capillary filled up with fluorescent solution (1), optofluidic connector (2), SMA905 optical fiber connector (3), mirror formed by a reflective sheet bonded onto a parabolic shaped 3D-printed base (4), fluorescence emission bandpass filter (5), sample solution inlet (6), housing (7), and O-ring seal of the capillary (8). Excitation light and sample inlet directions are illustrated by blue and yellow arrows, respectively.

**Figure 3 toxins-14-00866-f003:**
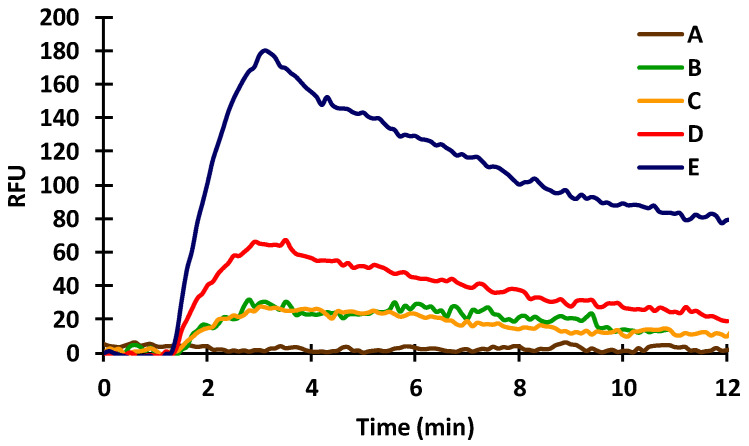
Effect of different concentrations of a-IgG–CF488A dye conjugate on the fluorescent signal in the presence of 10 µg/mL a-BSA at 0 µg/mL (A), 1 µg/mL (B,C), 2 µg/mL (D), and 5 µg/mL (E).

**Figure 4 toxins-14-00866-f004:**
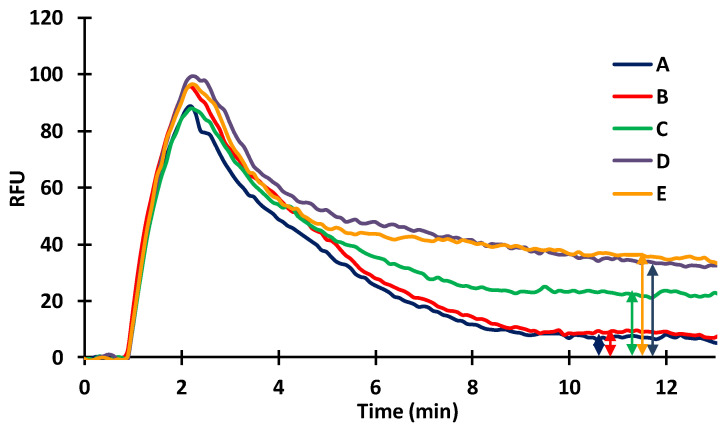
Effect of immobilized BSA on the immunoreactions of a-BSA (A,B: a-IgG–CF488A; C: a-BSA then a-IgG–CF488A injected consecutively; D: mixed a-BSA and a-IgG–CF488A injected; E: mixed and incubated a-BSA and a-IgG–CF488A at 31 °C for 3 min).

**Figure 5 toxins-14-00866-f005:**
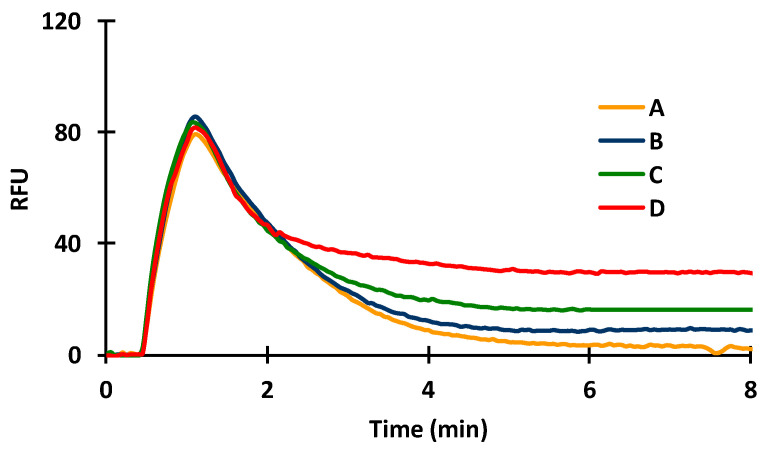
Effect of primary antibody concentration on sensor sensitivity (A: BSA only; B: 2 µg/mL a-BSA followed by the dye conjugate; C: 10 µg/mL a-BSA followed by the dye conjugate; D: 10 µg/mL a-BSA mixed with the dye conjugate).

**Figure 6 toxins-14-00866-f006:**
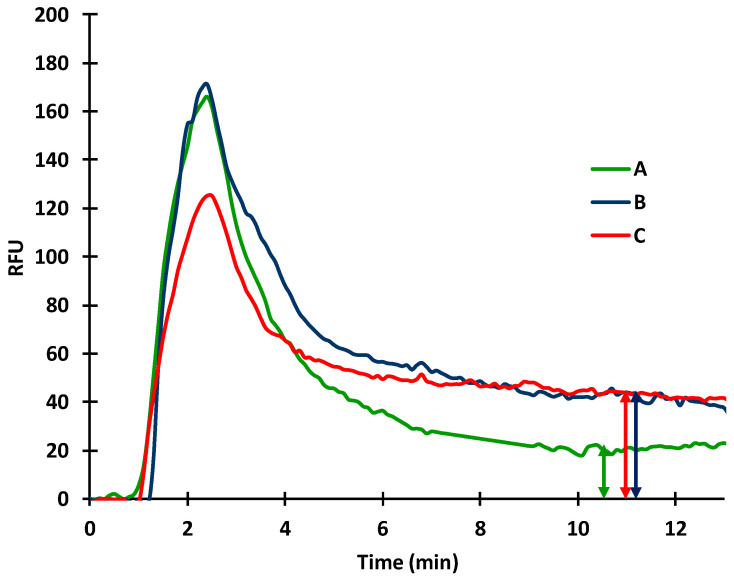
Effect of AcN extractant (6:4 AcN:water mixture) on the sensor response of BSA—a-BSA measurement (20 µg/mL BSA immobilized, 10 µg/mL a-BSA; A: 50% extractant; B: 25% extractant; C: 10% extractant).

**Figure 7 toxins-14-00866-f007:**
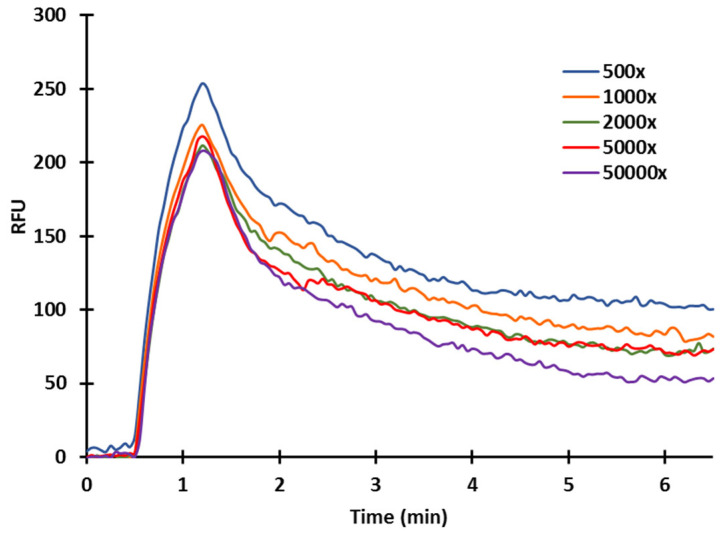
Examination of serum concentrations of a-ZON (at 20 µg/mL ZON-BSA conjugate immobilized).

**Figure 8 toxins-14-00866-f008:**
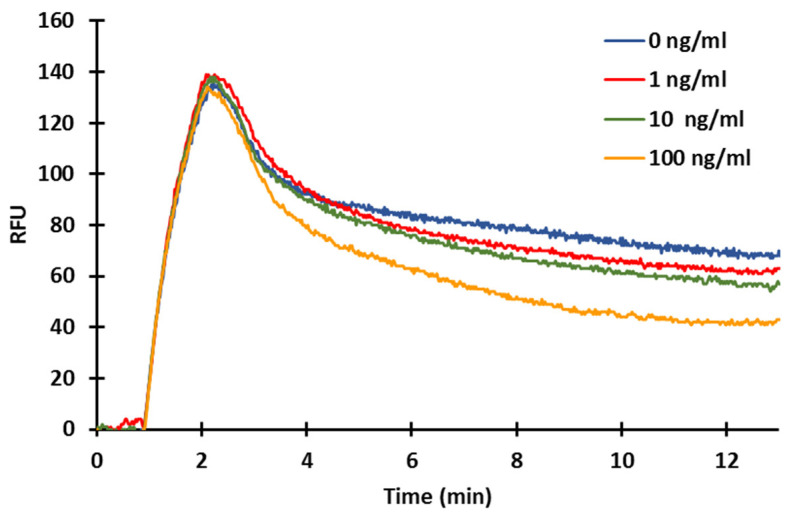
Signal curve for standards of ZON measurement (10 μg/mL ZON-BSA fixed, 2000×diluted serum).

**Figure 9 toxins-14-00866-f009:**
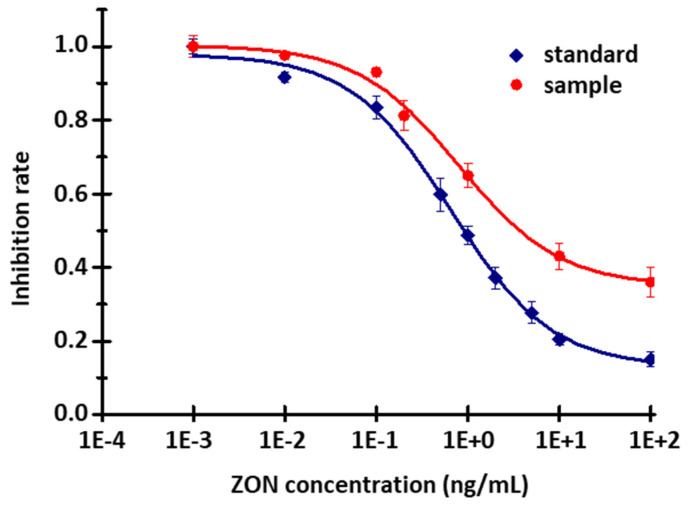
Calibration curve of ZON content of different diluted barley samples (10 μg/mL ZON-BSA fixed, 2000× diluted serum; standard—R^2^ = 0.97; EC_50_ = 0.68 ± 0.08 ng/mL; samples—R^2^ = 0.993; EC_50_ = 0.83 ± 0.12 ng/mL.

**Figure 10 toxins-14-00866-f010:**
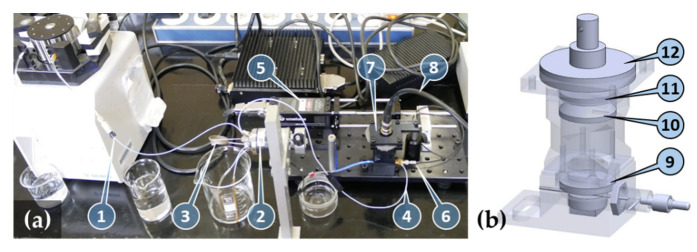
(**a**) Image of the complete system: pump with pump efflux (1), manual sample injector (2) with sample loop (3), capillary influx from injector (4), laser source (5), optical fiber connected to the optofluidic connector (6), optical measuring head (7), and detector output (8). Computer-controlled detector amplifier and signal processing electronics are not shown. (**b**) Transparent view of the optical measuring head: fluorescence emission filter (9), collimating lens (10), dichroic filter (11), and large area photodetector (12).

## Data Availability

The data presented in this study are available on request from the corresponding author. The data are not publicly available.

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
