# Peer review of "Development of an Immunofluorescent Capillary Sensor for the Detection of Zearalenone Mycotoxin"

_toxins, 2022, doi:10.3390/toxins14120866_

Round 1
Reviewer 1 Report
As authors mentioned, comprehensive detection techniques are required to monitor the ZON contamination in food and feed. This study aimed to develop a simple, low-cost device and assay platform based on A capillary immunofluorescence sensor for initial screening of ZON contamination. The manuscript was well prepared and it was interesting for readers of this journal. But there two main defects to constrain its integrity as an determination technique.
1.L285-295. Why was the ELISA method used as the reference method? There are so many commercial ELISA kits with very different accuracy and it was not fit as the reference method. The HPLC or HPLC-MS was recommended to evaluate your novel instrument and method.
2.More careful methodology validation was needed. The detail process should be supplemented in Materials and Methods, and the data results and the parameters such as the range of linearity, the recovery rate, the limit of detection and the limit of quantification should be presented.
Reviewer 2 Report
The manuscript describes the development of a sensor for the mycotoxin zearalenone (ZON). The sensor is based upon the binding of a fluorescently labeled secondary antibody to a ZON antibody (primary antibody) that has attached to a ZON-protein conjugate immobilized on the inner surface of a capillary. The novelty of the manuscript resides in the characteristics of the device, which will be of interest to those working on biosensors for detection of small molecules. The manuscript is generally well written. The quality of the experimental work appears to be good. Overall, this appears to be a nice bit of work, but it does have some issues that need to be addressed.
I had three general comments and many specific comments.
General Comments:
The manuscript describes results from some preliminary experiments where the sensor was developed to detect immobilized BSA. While this was clearly a step that the authors took in developing the ZON sensor, its presentation is not necessary. Section 2.4, which describes the effects of acetonitrile on the model (BSA) detection system is of particularly low relevance. The reason: the ZON system used a different antibody/antigen combination and therefore will very likely respond differently to acetonitrile. In addition, it would be much better if Figure 6 described the effects of acetonitrile on the ZON assay instead of the BSA model system. I recommend that the preliminary BSA work, including section 2.4 and Figure 6 be placed in a Supplemental section.
Although the lengths of some of the incubations are described, nowhere is it explicitly stated how long it takes to test a sample for ZON. This would be useful information for those interested in the technology.
The Materials and Methods section does not contain details on how the samples were tested, the length of incubation times, the concentrations of reagents, flow rates, etc. Some of this information could be found in the Results and Discussion section, but much of it was lacking. The specific comments deal with this in greater detail.
Specific Comments:
Abstract, lines 9-11 and 17-18. These lines describe methodological aspects of the manuscript. That is fine, except that what has been left out are results: specifically, a description of the basic performance parameters for the ZON assay: limit of detection (LOD), limit of quantification (LOQ), etc. I think these results would be of greater interest to the reader.
The authors hint at the ability to re-use the sensor. This could be a very important attribute of the sensor, but it does not appear to have been explored by the authors, or at least it hasn’t been described in the manuscript. If there is data on re-usability, it would be worthwhile to consider including it.
Introduction l.53: the statement “…they suffer from several limitations such as involvement of complicated instrumentation or being inexpedient for on-site application…” is clearly and demonstrably wrong. There are multiple commercial immunoassays (ELISAs and Lateral flow test strips) that are field portable and relatively inexpensive. I suggest including some additional citations, or links to the manufacturers of commercially available field-portable test kits.
Figures 1 and 2 were very helpful in understanding the configuration of the instrument.
l.132: What was the flow rate used? How long did washing out and stabilization take?
l.148: what concentration of BSA was used?
l.153-154: How long was the immunocomplex formation step? Was the flow stopped during this step?
l.155: How long was the washing step?
l.159: How long was the regeneration step with HCl?
General: at what time point were the measurements taken? From Figure 4 and Figure 6 it looks like the values were collected at a timepoint somewhere between 10 and 12 min after injection, but I did not see it specified in the text.
Figure 5: suggest “Effect of primary antibody concentration…”
Figure 8: Check the units used in the figure. They are ug/mL here, but ng/mL in Figure 9, which means they could be off by 1000-fold (?)
Figure 9: This figure is the culmination of much of the work presented in the manuscript. What do the error bars represent? How many replicates were used? Were they replicate samples or replicate injections of the same samples? Were the data collected on one day, or collected on multiple days?
l.294: suggest “…the ELISA reference method using the same anti-sera. The selectivity…”
l.298: I suggest removing the word “cheap” from the description here. There was no data presented on the costs of the assay compared to the costs of other methods, so there is no basis for saying it is less expensive.
Round 2
Reviewer 1 Report
Authors responded my comments well and I agreed it could be accepted in the present status.